# Isolation and Identification of Caprine Arthritis Encephalitis Virus from Animals in the Republic of Mordovia

**DOI:** 10.3390/ani13142290

**Published:** 2023-07-13

**Authors:** Olga Kolbasova, Timofey Sevskikh, Ilya Titov, Denis Kolbasov

**Affiliations:** Federal Research Center for Virology and Microbiology, Academician Bakoulov Street, Bldg. 1, 601125 Volginsky, Russia; titoffia@yandex.ru (I.T.); kolbasovdenis@gmail.com (D.K.)

**Keywords:** CAE, isolate, virus CAE, virus isolation, cell culture, sequencing, phylogenetical analysis

## Abstract

**Simple Summary:**

In the Russian Federation, small ruminant lentiviruses (SRLVs) circulate, causing slow infections that often develop and progress without clinical signs. The detection of seropositive animals leads to economic restrictions, including the prohibition of livestock product sales and the sale of breeding animals. In the investigation of biological samples from a clinically asymptomatic goat from a private farm, serological and molecular genetic methods (ELISA, PCR) yielded a positive result for lentivirus and antibodies against it. Subsequently, the virus was isolated in cell culture, and its presence was confirmed using transmission electron microscopy. Phylogenetic analysis revealed that the virus belongs to subtype B1. Extensive monitoring studies in different regions of the Russian Federation would give a clearer picture on the distribution of SLRVs and their classification.

**Abstract:**

This article presents the results of virological and genetic studies of an isolate of caprine arthritis encephalitis (CAE) virus from the republic of Mordovia, Russian Federation. The isolate was found during monitoring studies of goat blood samples for the viral genome, and the presence of antibodies to lentiviruses was detected. According to the recommendation of the OIE, the positive result of PCR was confirmed with nucleotide sequencing. It was found that the obtained nucleotide sequence is identical to the genome of small ruminant lentiviruses presented in the GenBank database. Phylogenetic analysis showed that the isolate “Mordovia-2018” was included in the same cluster with an isolate from the Tver region of the Russian Federation detected in 2008. The sequence of the fragment of the *env*-gene of the isolate from the republic of Mordovia is available in GenBank under the number MN186380.1. To isolate the virus, a fraction of peripheral blood monocyte cells from the animal’s blood was added to a monolayer of lamb synovial membrane cell culture, and ten passages were carried out. The first manifestations of the cytopathic effect were observed after the third passage on the eighth day of cultivation in the form of single large cells of irregular shape with 5–7 nuclei. At the seventh passage, multiple syncytium with 7–12 nuclei were observed. At subsequent passage levels, the formation of syncytium containing more than 10–14 nuclei was observed.

## 1. Introduction

Small ruminant lentiviruses (SRLVs) include maedi visna virus (MVV) and caprine arthritis encephalitis virus (CAEV). SRLV disease is associated with progressive and persistent inflammatory lesions in organs, including the lungs, joints, udder and central nervous system. Mastitis is common in both host species; pneumonia is the main clinical sign in sheep, while arthritis is the prominent feature in clinically affected goats. Caprine arthritis encephalitis (CAE), also known as caprine leukoencephalomyelitis arthritis, is a slow viral disease characterized mainly by progressive arthritis; the clinical symptoms may also include a syndrome of demyelinating encephalitis, interstitial pneumonia and intralobular mastitis [1,2,3].

Goats of all ages are susceptible to SRLVs. The incubation period can range from one month to several years. CAE is characterized by lifelong persistence of the pathogen in monocytes and macrophages of the host [4,5,6] as well as variability in the incubation period and the induction of serologically detectable antiviral antibodies. CAE infection has been reported worldwide and is highly prevalent in countries that practice intensive production of dairy goats, such as Australia, USA, Canada, Japan, Norway, France and Switzerland [7].

SRLVs cause both direct and indirect economic losses owing to decreased productivity and delayed maturity [8].

CAE is characterized by respiratory, nervous, mammary and joint clinical signs. The clinical lesion appears to depend on the tropism of the SRLV strain, the affected species and the genetic background of each breed or animal. Although the process usually proceeds subclinically, a small percentage of animals may present some or all of these signs [9].

The main risk factor for the transmission of SRLVs within a herd is ingestion of infected colostrum/milk or direct contact with infected animals [3]. In addition, large herd size, increased age, high stocking density, prolonged duration of exposure to infected animals and extensive rearing systems are also indicated as risk factors [10,11].

The causative agent of CAE belongs to the genus *Lentivirus* of the *Retroviridae* family. Retroviruses are a family of complex RNA viruses that form a DNA copy of the genome using reverse transcriptase and integrate this copy into the host genome causing latent infection [12,13,14]. The MVV and CAEV genomes range in length from 8400 to 10,000 nucleotides (nts) and consist of three main genes common to all replication-competent retroviruses, *gag*, *pol* and *env*, and several regulatory genes. The proviral DNA is flanked by repeating sequences known as long terminal repeats (LTRs), which contain promoter elements that initiate the DNA transcription of and play an important role in cellular tropism and in pathogenesis [15].

According to phylogenetic studies, SRLVs are currently divided into five genetic groups from A to E [16]. Subtypes A5–A7 and genotypes C and D circulate exclusively in goats; the subtype A2 was detected only in sheep; subtypes A1, A3, A4, A6, B1 and B2 were found in both sheep and goats [17]. The nucleotide sequences between genotypes differ by 25–37%.

In addition, genotypes A, B and E contain subtypes whose sequences differ by 15–27%. According to the actual data, genotype A has twenty-seven recognized subtypes (A1–A27) [18], genotype B has five subtypes (B1–B5) [19], and genotype E has two identified subtypes (E1 и E2) [20,21].

Most infected goats do not have clinical signs of the disease but remain seropositive and are the source of the virus [3,10,22,23,24,25,26].

CAE is registered worldwide and causes economic losses in the affected countries [12,27,28]. Numerous studies describe the phenomenon of interspecies transmission between goats and sheep, which can stimulate the emergence of new strains that are possibly capable of displaying other biological properties [22,29,30,31,32].

CAE is not characterized by seasonality, periodicity of epizootics or geographical timing. CAE is a serious concern for animal husbandry due to its latent course, fatal outcome and lack of treatment and prevention.

In 2011, we studied and established a trend towards transgression of the nosoarea and an increase in epizootic tension for this disease in the territory of the Russian Federation [33]. According to the results of serological studies at the Federal Research Center for Virology and Microbiology (FRCVM), many regions of Russia are affected by CAE (Figure 1).

The lack of a functional system of prevention, monitoring and laboratory diagnostics of slow infections combined with the wide geographical distribution of CAE can lead to the unfavorable development of the epizootic situation in the country and expansion of the nosoarea.

The use of coordinated diagnostic and monitoring methods as well as a detailed study of the pathogens of slow infections circulating in Russia are important for the development of a set of preventive measures. There is currently not enough information on the prevalence of SRLVs in Russia and their genetic characteristics, so it is important to conduct monitoring studies, test different diagnostic approaches and genotype discovered isolates. Therefore, the purpose of our study was to test different diagnostic approaches, including virus isolation in cell culture and genetic characterization of the isolated virus.

## 2. Materials and Methods

### 2.1. Samples

The studied samples were the blood and serum of a pregnant, clinically healthy 2-year-old goat obtained during routine veterinary monitoring on a private farm and delivered to the FRCVM Testing Center from Mordovia in 2018. This study did not involve humans or animals.

### 2.2. Extraction of DNA

DNA was extracted using a commercial QIAamp^®^ DNA Mini Kit (QIAGEN) according to the manufacturer’s instructions.

### 2.3. PCR

PCR was performed in accordance with the guidelines for the detection of proviral DNA of small ruminant lentiviruses with polymerase chain reaction [34]. The following primer sequences flanking the *env*-gene region were used:VVM-F 5′-AYARAAATKAGAGYAACAAGTGGAC-3′VVM-R 5′- CCATATGTKTAAGCTCTTACMYTWATTACTT-3′

A master mix consisting of 11 µL of RNase-free water, 1 µL of each primer (10 µM), 5 µL of 5× PCR buffer, 0.5 µL of 25 mM dNTPs, 1 µL of 25 mM MgCl2, 5 µL of DNA template and 1.0 U of Taq-polymerase.

The PCR conditions were set as follows: polymerase activation at 95 °C for 3 min. This was followed by 30 cycles subdivided as follows: denaturation at 94 °C for 15 s, annealing at 50 °C for 15 s, elongation at 72 °C for 15 s and a final extension at 72 °C for 5 min. The PCR products were resolved with electrophoresis on a 2.0% agarose gel.

### 2.4. Serological Studies

The detection of antibodies to the CAE virus in the serum was carried out using a commercial ID Screen^®^ MVV/CAEV Indirect—Screening test (Indirect ELISA for the detection of antibodies against MVV/CAEV in sheep and goat serum, plasma or milk) (IDVET, Grabels, France) according to the manufacturer’s instructions.

### 2.5. Isolation of Peripheral Blood Monocyte Cells (PBMC) Fraction

PCR-positive blood was diluted 2 times with sterile buffered saline solution and transferred to a 50 cm^3^ centrifuge tube. A total of 10 mL of sterile Ficoll solution (density 1077 g/cm^3^) was layered under the diluted blood using a syringe with a long needle. The mixture was centrifuged at 1500 rpm for 20 min.

Then, a fraction of PBMC was taken and washed twice with buffered saline solution. The cells were resuspended in 10 mL of DMEM medium.

### 2.6. Cultivation and Infection of Lamb Synovial Membrane (LSM) Cells

LSM cells from the FRCVM cell culture bank were cultivated to a stationary monolayer in DMEM medium with 7% FBS. The culture was in contact with viral material for 2 h at 37 °C. Then, the cell monolayer was washed with medium, and a maintenance DMEM medium with 2% fetal cattle serum and a complex of antibiotics (penicillin–streptomycin 10,000 U/mL, 10,000 mg/mL, respectively, and 1% Amphotericin B 250 mg/mL) was added.

An intact cell culture served as a control. The cells were incubated in a CO2 incubator at 37 °C until the most extreme cytopathic effect was manifested (from 7 to 14 days).

### 2.7. Nucleotide Sequencing and Phylogenetic Analysis

Nucleotide sequencing of the *env*-gene of the CAE virus was performed using type-specific primers selected for the virus detection with PCR.

The PCR products were purified from the gel using a commercial PCR Purification Kit QIAquick (QIAGEN, Stockach, Germany) according to the manufacturer’s instructions.

Sequencing was performed using the Big Dye Terminator kit 3.1 (Applied Biosystems, Waltham, MA, USA) on the Applied Biosystems 3130 Genetic Analyzer (Applied Biosystems, Waltham, MA, USA).

The obtained sequences were compared with the *env*-gene sequences published in the GenBank database. Alignment was performed using the BioEdit program [35]. The MEGA program was used to construct a phylogenetic dendrogram [36]. Reference genome-wide sequences deposited in the Genbank database under the numbers AY900630; GU120138; M60610; M10608; L06906; EU293537; GQ381130; JF502416; JF502417; HM210570; AF322109; AF479638; FJ195346; HQ848062; AY445885; M33677; U64439; MN186380.1 and JN008914.1 were used in the construction of the dendrogram.

### 2.8. Transmission Electron Microscopy

Transfected or infected LSM cells were detached from the culture flask with trypsin–EDTA treatment (Life Technologies, Waltham, MA, USA), diluted with PBS and centrifuged at 1500× *g* for 10 min at 4 °C. The cell pellets were resuspended in Sorensen’s buffer (0.1 M NaPO4, pH 7.4) and, for structural observation, fixed with 0.5% glutaraldehyde/1.6% paraformaldehyde (Sigma, Darmstadt, Germany), postfixed in 1% OsO4, dehydrated in ethanol and included in Epon resin. Thin sections were performed with an ultratome MTX (RMC) and observations were made using a Tecnai Spirit electron microscope (FEI Company, Hillsboro, OR, USA).

## 3. Results

Samples of blood and serum from one animal were studied. For the diagnosis of CAE, an integrated approach was used, including classical (serological methods, isolation and cultivation of the virus in permissive cell culture) and molecular genetic research methods (PCR and sequencing). The serum was studied with ELISA; the blood was used both for PCR testing and isolation of the virus in cell culture.

To detect the CAEV in the blood sample from the animal, total DNA was extracted, and a fragment of the *env*-gene with a length of 213 bp of the CAEV was amplified (Figure 2).

When analyzing the blood serum sample using a commercial ELISA kit, antibodies to small ruminant lentiviruses were detected. The test showed that the SP% value was 352, which significantly exceeded the positive threshold of 60.

To isolate the virus from the blood of the infected animal, a fraction of PBMC was isolated. The resulting fraction of PBMC was introduced to the monolayer of LSM cells and cultivated until signs of cytopathic effect caused by the virus infection were observed. No signs of cytopathic virus action were observed on the permissive subculture of lamb synovial membrane cells after two passages. The first manifestations of the cytopathic effect were noticed at the third passage on the eighth day of cultivation by the appearance of single large cells with an abnormal shape and five to seven nuclei (Figure 3A,B). However, the number of affected cells in the culture was minimal (only four to five cells per 25 cm^2^).

At the seventh passage on the fifth to seventh day of cultivation, numerous multinucleated cells with seven to fourteen nuclei were observed (Figure 4).

To observe the viral particles, transmission electron microscopy was performed on positively stained samples of the infected cell culture. C-type particles, typical for CAEV, were observed (Figure 5).

To determine the molecular genetic characteristics of the isolated CAE virus, nucleotide sequencing of the *env*-gene fragment was performed. The obtained primary nucleotide sequence of the *env*-gene fragment was used for phylogenetic analysis and dendrogram construction (Figure 6).

The presented dendrogram shows that the studied isolate “Mordovia/2018” is part of a cluster formed by strains from China, the USA and Mexico belonging to genotype B1, as is the isolate “Tverskoy/2008”. It is possible that the origin of these two isolates is common. At the same time, the length of the analyzed sequence is too small to make any specific conclusions.

## 4. Discussion

The causative agent of CAE induces slow infections with a latent course, which can lead to death. The incubation period ranges from one month to several years. SRLVs spread through many ways, including close contact between healthy and infected animals, the environment and fomites and feeding with infected milk. Many infected animals remain asymptomatic carriers that constantly release the virus into the environment. There are no specific measures for the treatment and prevention of this disease.

The CAE virus shows tropism mainly to monocytes and macrophages [37], and infected cells localized in the red bone marrow act as a reservoir of the virus in the body. In cells, the virus persists in the form of proviral DNA [38,39].

It is hard to detect the disease and estimate the prevalence of lentiviruses due to the often-asymptomatic course of infection progression, which further contributes to the spread of the virus and development of the disease. Since CAE leads to significant economic damage as a result of trade restrictions, it is necessary to swiftly prevent the spread of infection. The most effective approach to solving this issue is routine serological screening to identify infected individuals.

ELISAs are widely exploited in SRLV control programs for the screening of goat and sheep populations. An advantage of ELISAs is the capability to be applied in various biological samples, such as blood serum, plasma and milk. ELISA remains a low-cost, user-friendly diagnostic test with sufficient repeatability, sensitivity and specificity, but its performance is not universally constant.

The usage of an ELISA and PCR combination allows for a precise diagnosis, which is especially important during the low seroconversion period. SRLVs show high genetic variability, which makes PCR a less reliable method [40].

To this end, it would be appropriate to draw the attention of owners who still underestimate this serious problem to a proper assessment of the state of animal health [41]. Serological screening together with a rigorous evaluation of the flock may greatly reduce the risk of infection.

The advantage of PCR compared to the serological methods is the early detection of the SRLV infection preceding the production of antibodies, which may occur months later.

The first successful PCR protocol applied for the detection of CAEV and MVV was developed by Zanoni et al. [42].

This led to more complex and reliable molecular diagnostic protocols. These were developed with other PCR techniques to improve the sensitivity, specificity and accuracy of molecular diagnostics (multiplex PCRs, (semi-)nested PCRs and real-time PCRs) but have been exploited with contradictory results.

A low viral load in cases of natural infection and the high genomic variability of SRLV greatly complicate the design of PCR tools and lower the chances of virus detection using only molecular diagnostic methods [43].

The isolation of viral DNA from an animal’s blood is problematic since PCR can detect it only at the early stages of infection or when the animal shows clinical signs of infection.

The diagnosis of SRLV is often hampered by the fact that it is impossible to amplify all samples with PCR, even though all samples were taken from seropositive animals. This may be due to the low recovery of template DNA since SRLV infects monocytes that do not all contain genetic material [40].

The situation with the identification and differentiation of pathogens is complicated by the imperfection of existing serological tests. As shown in Schaer J. et al. 2022, when comparing different serological tests, there are often some differences in the results obtained. At the same time, typing of SRLVs with serological methods is inferior to PCR with subsequent sequencing of the PCR products.

As a result, the authors recommend using at least two serological tests for diagnosis, and in turn, differentiation and genotyping should be performed with PCR followed by sequencing [43].

In the Russian Federation, many regions are affected by CAE, but official data may not reflect the real situation for this nosounit, since laboratory studies were conducted only for a limited number of herds, and there is no global monitoring system for CAE. This fact is confirmed by local studies of the persistence of CAEV on small farms. Thus, a study by Shuralev E. et al. indicates that CAEV is present and circulates on small amateur goat farms in the Republic of Tatarstan (Russia), and it currently remains unnoticed in the absence of a control or monitoring program [44].

In this paper, the SRLV strain was found in naturally infected goat flocks. Analysis of the obtained samples with PCR and ELISA showed the presence of a virus, but during virological studies, no changes were observed in the cellular monolayer after two passages.

On the third passage, single cells with five to eight nuclei were detected on the eighth day after infection; however, the number of infected cells in the culture flask was minimal. The cytopathic effect characteristic of lentiviruses (the phenomenon of formation of syncytium when fused cells contain from three or four to dozens of nuclei and have variable size) was observed only after the seventh consecutive passage.

Phylogenetic analysis of the sequences of the *env*-gene from the SRLV isolated in this study revealed that this isolate belonged to the SRLV subtype B1 according to the classification by Rolland et al. [45].

Sequences for the *env*-gene available in Genbank were used to construct the dendrogram. It should be noted that only two sequences of small ruminant lentiviruses isolated in the Russian Federation are available in Genbank (MN186380.1—Mordovia 2018; JN008914.1—Tverskoy 2008). Due to the high variability of the genomes of lentiviruses, their phylogenetic analysis is complicated [46].

Sequencing of the *env*-gene of the studied isolate showed it belonging to the B1 genotype. The previously studied isolate “Tverskoy/2008” belongs to the same genogroup, so they may have the same origin.

## 5. Conclusions

We isolated the CAE virus on the lamb synovial membrane cell culture and named the isolate “Mordovia/2018”. The detection of the presence of CAEV in samples from goat farms located in various often very remote regions may indicate the widespread distribution of CAEV in the Russian Federation.

In this regard, it is necessary to carry out extensive monitoring activities based on serological and molecular genetic methods to obtain a clear picture of the distribution and classification of CAEV.

## Figures and Tables

**Figure 1 animals-13-02290-f001:**
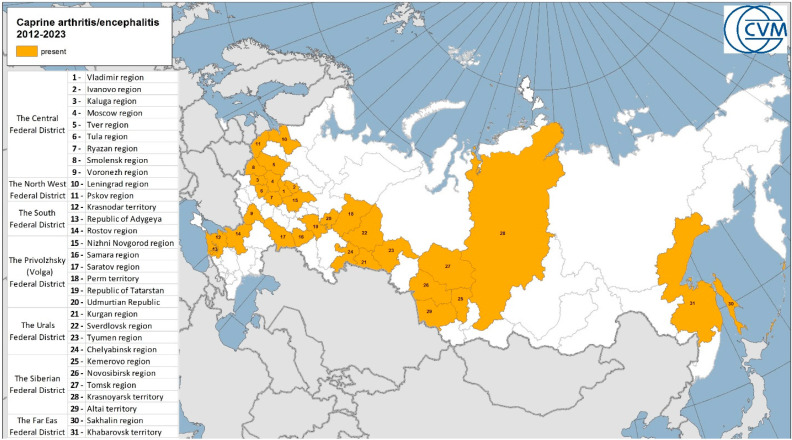
Registration of caprine arthritis encephalitis cases (seropositive animals) in the Russian Federation (FRCVM unpublished data).

**Figure 2 animals-13-02290-f002:**
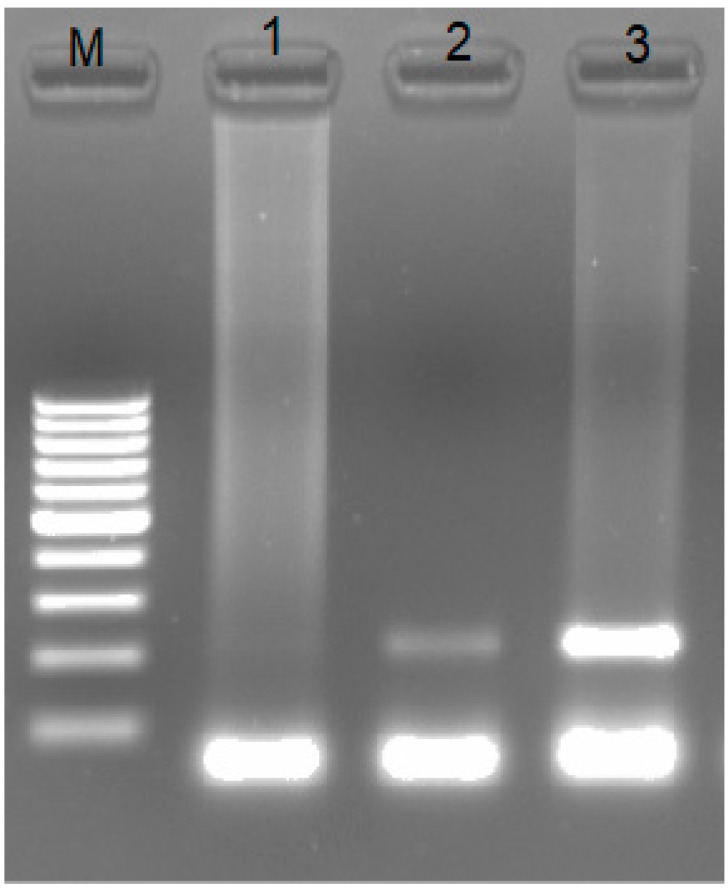
Results of detection of the lentivirus genome: 1—negative extraction control; 2—the test sample; 3—positive PCR control; M—Nucleic Acid Molecular Weight Marker 100 b.p.—1 kb. Fragments below 100 b.p. are primer dimers.

**Figure 3 animals-13-02290-f003:**
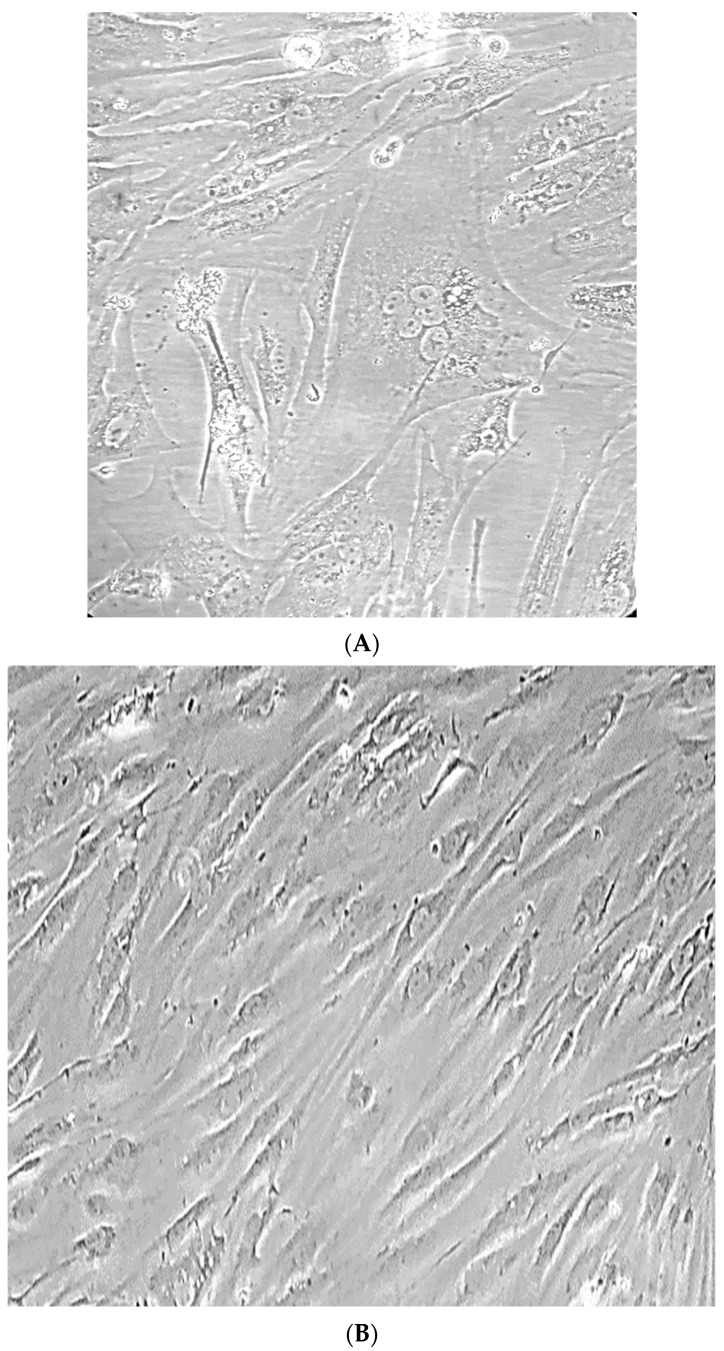
(**A**) Monolayer culture of infected cells on the 8th day after infection, 3rd passage (native sample ×380). Arrows show large cells with 5–7 nuclei; (**B**) The LSM culture control (×380).

**Figure 4 animals-13-02290-f004:**
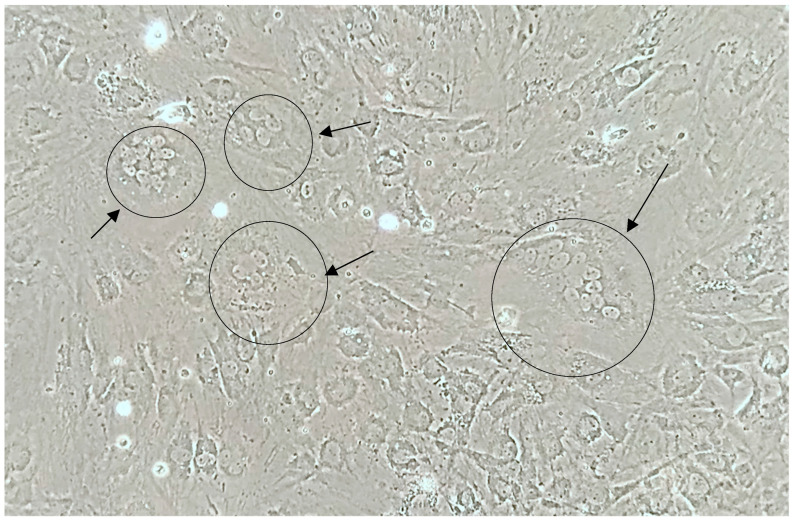
Monolayer culture of synovial membrane cells of infected lamb on the 7th day after infection (7th passage, native sample ×380). Arrows show multinucleated cells.

**Figure 5 animals-13-02290-f005:**
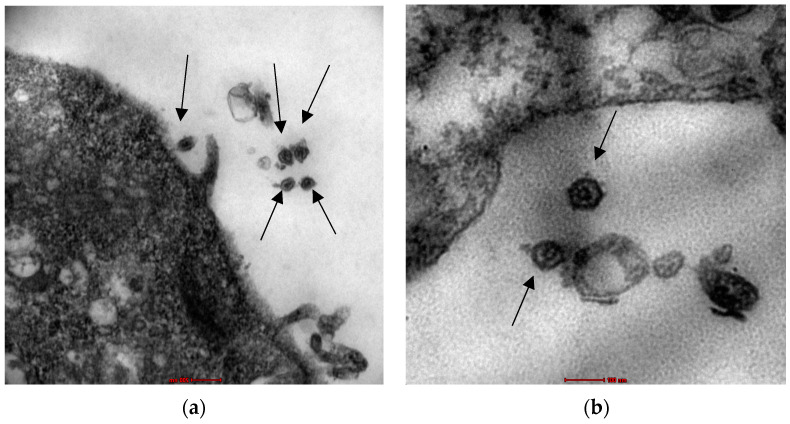
Transmission electron microscopy on CAEV-infected LSM cells. (**a**) Magnification 37,000×; (**b**) Magnification 97,000×. Arrows point to SRLVs virions.

**Figure 6 animals-13-02290-f006:**
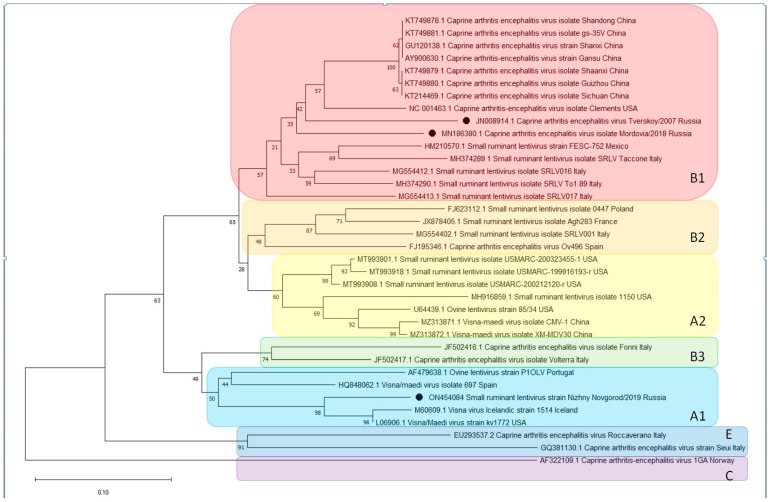
A phylogenetic dendrogram constructed on a fragment of the *env*-gene with the neighbor joint method with bootstrap = 1000. The letters (A1, A2, B1–B3, C, E) indicate genotypes usually identified with phylogenetic analysis of lentiviruses.

## Data Availability

The data presented in this study are available on request from the corresponding author.

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
