# Peer review of "Isolation and Identification of Caprine Arthritis Encephalitis Virus from Animals in the Republic of Mordovia"

_animals, 2023, doi:10.3390/ani13142290_

Round 1
Reviewer 1 Report
Kolbasova et al. present detection and phylogenetic analysis of small ruminant lentivirus (SRLV) from Mordovia. They also show cytopathic effect after cocultivation of infected cells.
Comments:
1. Fig 1 - map unclear. The numbered regions need to be shown on the map. Otherwise the figure is unusable.
2. At the beginning of results, it should be clearly stated how many samples were analyzed. Also, which were analyzed by serology and which by PCR
3. Figure 2 - A gel elfo is shown. What does the horizontal arrow mean? This should be explained in legend. There is huge band in negative control, most probably primer dimers. This should be mentioned
4. Fig 3 Showing fractionated blood is in my opinion unbecessary and this figure should be deleted
5. Figure 7 - the authors claim that the current Monrovia isolate belongs to B1 genotype. Hower, based on the NJ tree shown, there is no good support to this. The tree nodes that cluster this sequence into B1 have weak suport (not even shown on the branches). This conclusion should be excluded, or more phylogenetic analysis should be added. E.g. seuqnecing other parts of the viral genome.
6. The authors cite a publication from 1998 as a reference for SRLV phylogeny (reg #46). I would suggest using one of several recent authoritative works, e.g. PMID: 36793939.
Author Response
Response to reviewer points.
Thank you for your valuable time. We revised document according to your suggestions.
- We changed the map in revised document.
- We added information on samples in the beginning of results section.
- We clarified the Fig. 2 legend and changed the figure itself.
- We removed this photo.
- We rearranged the dendrogram using more sequences from Genbank.
- Thank you for this reference, we added it to the list.
Reviewer 2 Report
The Authors present an interesting study on the Isolation and identification of caprine arthritis encephalitis virus from animals in the Republic of Mordovia. The manuscript is well written but few clarifications are required to make this study complete.
Line 74: interspecific transmission? please check
Line 82: FRCVM, expand the short form for the first time.
Introduction: Specify aim/importance of this study eventhough under the section on discussion, authors provided a clear explanation.
Line 97, 2.1 Samples- 'during monitoring' please specify.
Line 126: These were peripheral blood monocyte cells or lymphocytes (PBMC)?
Line 129: LSM cells's source?
Line 134: would be ideal to include incubation dates
Line 168: Name of the ELISA kit used
Line 185, Figure 4- Please use the same magnification for the images of the virus-infected cells and the control cells.
Author Response
Response to reviewers report.
Thank you for your valuable time! We modified manuscript according to your suggestions.
- "Interspecific" changed to "interspecies".
- We expanded abbreviation in the text.
- We added a sentence in our introduction specifiying the importance and aim of our study.
- We spicified it in revised manuscript, smaples were taken during routine veterinary monitoring for SRLVs.
- We meant peripheral blood monocyte cells (PBMC). We changed it in manuscript.
- Cell culture bank of Federal Research Center for Virology and Microbiology, added in manuscript.
- We specified period of incubation.
- Full name ELISA kit is included in M&M's section.
- We replaced photos, now they are under the same magnification.
Reviewer 3 Report
Abstract
Line 10: The word detection should be add to the end of the “antibodies to lentivirus”
Line 15: Tver región “from Russia feretation should be add
Line 20: The word symplasts should be change to Syncytium
Introduction
Line 36: Goats are susceptible to the Virus infection, not susceptible to this disease. The disease is the cause of the virus infection.
Materials and methods
Line 99: FRCVM?
Line 101: The word Isolated should be remplaced for the word “Extraction”
Line 129: LSM are from primary culture?, if is the origen of the culture should be mentioned how this were obtained.
Line 149-150: Between both line a space should be add
Results
Fom line 162 to 164: The paragraph should be modified to a better commpresion
Line 171: Lymphocytes and monocytes cell should be remplaced to Leucocytes.
Line 177: The paragraph “until signs of cytopathic effect of the virus appeared” should be modified. I recomend for example "until signs of cytopathic effect caused to the virus infection”.
Figure 4 and 5: Is dificult to see the multinuclested cello r the syncytia. I recomend stain the monolayed to aprresiatte the Cytophatic effect
Figure 6: Arrows or circles should be used to identify the virions
Discussion
Line 269: The authors should be explain why is recommended the use of both diagnostic test (molecular and serology) to detect SRLV. It is due to the variability on the secuence of this lentivirus. I considere that is phrase should be included.
English need to be corrected on some sentences in the manuscript
Author Response
Thank you for your valuable time and input in the quality of our manuscript!
Abstract:
- We included word "detection", I guess we lost it during translation somehow.
- Russian Federation added.
- We switched word symplasts to syncytium.
Introduction:
- You are correct, we changed the sentence.
M&M's:
- We wrote the full name of organisation in introduction section.
- We agree, changed it in manucript.
- We specified the source of cell culture. It was taken from biobank of cell cultures (FRCVM, Russia).
- Thank you, we added a space.
Results.
- We modified the sentence.
- According to reccomendation from other reviewer, we canged any mention of leucocytes and monocytes fraction to PBMC (peripheral blood monocyte cells).
- We followed your reccomendation.
- We increased sharpness and contrast of images for better visualisation. Unfortunately, we won't be able to make better photos.
- We added arrwos to identify the virions.
Discussion.
1. We explained it earlier in the text (lines 298-300 in revised manuscript).
Round 2
Reviewer 1 Report
My comments have been answered by the authors
Reviewer 3 Report
The manuscript was improved for publication